# The Role of Autophagy and lncRNAs in the Maintenance of Cancer Stem Cells

**DOI:** 10.3390/cancers13061239

**Published:** 2021-03-11

**Authors:** Leila Jahangiri, Tala Ishola, Perla Pucci, Ricky M. Trigg, Joao Pereira, John A. Williams, Megan L. Cavanagh, Georgios V. Gkoutos, Loukia Tsaprouni, Suzanne D. Turner

**Affiliations:** 1Department of Life Sciences, Birmingham City University, Birmingham B15 3TN, UK; tala.ishola@bcu.ac.uk (T.I.); megan.cavanagh@mail.bcu.ac.uk (M.L.C.); loukia.tsaprouni@bcu.ac.uk (L.T.); 2Division of Cellular and Molecular Pathology, Department of Pathology, University of Cambridge, Cambridge CB2 0QQ, UK; pp504@cam.ac.uk (P.P.); ricky.m.trigg@gsk.com (R.M.T.); sdt36@cam.ac.uk (S.D.T.); 3Department of Functional Genomics, GlaxoSmithKline, Stevenage SG1 2NY, UK; 4Department of Neurology, Massachusetts General Hospital, Harvard Medical School, Boston, MA 02114, USA; jtavaresdasilvapereira@mgh.harvard.edu; 5Institute of Translational Medicine, University Hospitals Birmingham NHS Foundation Trust, Birmingham B15 2TH, UK; j.a.williams@bham.ac.uk; 6Institute of Cancer and Genomic Sciences, College of Medical and Dental Sciences, University of Birmingham, Birmingham B15 2SY, UK; 7Mammalian Genetics Unit, Medical Research Council Harwell Institute, Oxfordshire OX110RD, UK; 8MRC Health Data Research Midlands, University of Birmingham, Birmingham B15 2TT, UK; 9NIHR Experimental Cancer Medicine Centre, Birmingham B15 2TT, UK; 10NIHR Surgical Reconstruction and Microbiology Research Centre, Birmingham B15 2TT, UK; 11NIHR Biomedical Research Centre, Birmingham B15 2TT, UK; 12Central European Institute of Technology (CEITEC), Masaryk University, 625 00 Brno, Czech Republic

**Keywords:** cancer stem cells (CSCs), tumour microenvironment, solid cancers, haematological malignancies, autophagy, LncRNAs

## Abstract

**Simple Summary:**

Cancer stem cells (CSCs) represent a distinct cancer subpopulation that can influence the tumour microenvironment, in addition to cancer progression and relapse. A multitude of factors including CSC properties, long noncoding RNAs (lncRNAs), and autophagy play pivotal roles in maintaining CSCs. We discuss the methods of detection of CSCs and how our knowledge of regulatory and cellular processes, and their interaction with the microenvironment, may lead to more effective targeting of these cells. Autophagy and lncRNAs can regulate several cellular functions, thereby promoting stemness factors and CSC properties, hence understanding this triangle and its associated signalling networks can lead to enhanced therapy response, while paving the way for the development of novel therapeutic approaches.

**Abstract:**

Cancer stem cells (CSCs) possess properties such as self-renewal, resistance to apoptotic cues, quiescence, and DNA-damage repair capacity. Moreover, CSCs strongly influence the tumour microenvironment (TME) and may account for cancer progression, recurrence, and relapse. CSCs represent a distinct subpopulation in tumours and the detection, characterisation, and understanding of the regulatory landscape and cellular processes that govern their maintenance may pave the way to improving prognosis, selective targeted therapy, and therapy outcomes. In this review, we have discussed the characteristics of CSCs identified in various cancer types and the role of autophagy and long noncoding RNAs (lncRNAs) in maintaining the homeostasis of CSCs. Further, we have discussed methods to detect CSCs and strategies for treatment and relapse, taking into account the requirement to inhibit CSC growth and survival within the complex backdrop of cellular processes, microenvironmental interactions, and regulatory networks associated with cancer. Finally, we critique the computationally reinforced triangle of factors inclusive of CSC properties, the process of autophagy, and lncRNA and their associated networks with respect to hypoxia, epithelial-to-mesenchymal transition (EMT), and signalling pathways.

## 1. Introduction

The process of cellular transformation resulting from the accumulation of mutations and outgrowth of cells in an evolutionary manner, according to fitness, is a central dogma of cancer biology. This stochastic process has been employed to explain intra- and inter-tumour heterogeneity. However, a second model exists whereby cancer stem cells (CSCs) represent the apex of a hierarchy from which progeny differentiate that then constitute the majority of the tumour mass. The two models are not necessarily mutually exclusive and may cooperate whereby CSCs may also evolve stochastically producing genetically diverse offspring [1]. Indeed, the stochastic clonal evolution model posits that tumour cells possess similar growth potential and through selective pressure, some sub-clones may become dominant constituting the majority of the tumour and further promoting growth [2,3]. Mounting evidence in the field of cancer biology suggests that CSCs exist in a variety of malignancies although with some caveats [4]. The CSC model states that not all cells of the tumour population possess similar potential, rather populations of cells can initiate tumours and possess properties such as quiescence and unlimited proliferative capacity [5,6]. Analogous to normal tissue ontogeny and stem cell development, CSCs in this model are proposed to be organised in a hierarchical manner [5]. Based on this model, CSCs sit at the top of the hierarchical order within the tumour cell population and govern tumour progression. In addition to unlimited proliferative capacity, CSCs can sustain tumour growth and maintenance and have been shown to form tumours in non-obese diabetic/severe combined immunodeficient (NOD/SCID) mice [5,7]. The process of self-renewal in CSCs, similar to stem cells, may be influenced by the microenvironment of the tumour and external factors, and hence they are not self-autonomous units [8]. Furthermore, CSCs express cellular efflux pumps, anti-apoptotic proteins, and have low levels of reactive oxygen species (ROS) [9]. Despite progress made in the field of cancer biology, the CSC model is still currently a subject of much debate. Nonetheless, in practical terms, cancer growth promotion, aggressive behaviour, and metastasis depend on the presence of these cells in cancers [10,11].

With respect to the origin of CSCs, they may originate from transformed tissue-specific stem cells [12,13]. However, it is also possible that CSCs arise from mutant progenitor cells that have acquired stem-like properties in response to the acquisition of certain genomic alterations [14]. CSCs were first identified in haematological malignancies and some solid cancers, and were thought to account for recurrence, metastasis, drug and radiotherapy resistance [15,16,17]. If the consensus definition of a CSC is a cell with self-renewal capacity that can give rise to a heterogeneous population within the tumour [14], then naturally this definition can encapsulate several cell populations. In many cases, the term CSC is used interchangeably with cancer-initiating cells and tumour-propagating cells. However, cancer-initiating cells are perhaps more representative of the cell of origin rather than the cell that exists in the established malignancy and drives tumour growth. Likewise, CSCs need not necessarily be transformed tissue-specific stem cells in which case, tumour-propagating cells are maybe more representative of the tumour cell population under discussion in this review (Figure 1). Another terminology that is commonly used is that of cancer persister cells which are a fraction of the tumour population that escape therapy and may remain dormant in niches, hence not being eliminated by immune surveillance or cancer therapy [18]. As such, drug resistance may be due to persister cells that manipulate their microenvironment and may undergo genetic and epigenetic changes that lead to drug resistance [19]. In this review, we will use the term CSC to describe the population of cells that drives tumour growth in the established malignancy and gives rise to the bulk tumour mass [20]. In experimental terms, this population is represented by the cells that generate a tumour that fully phenocopies the primary tumour by serial xenotransplantation through immunocompromised mice or via sphere-forming assays in vitro [5]. In essence, CSCs, like tissue-specific stem cells, are considered relatively quiescent, represent the minority of the tumour mass, can self-renew, and give rise to progeny. However, the notion that CSCs constitute a minor population was challenged by a study of melanoma in which approximately 25% of unselected cells from primary and metastatic melanomas formed tumours in immunodeficient mouse models [21,22]. Moreover, accumulating evidence implicates long noncoding RNAs (lncRNAs) in regulating key aspects of CSC properties such as the maintenance of stemness, self-renewal, and tumour progression [23,24,25]. For instance, *H19* is involved in tumourigenesis and cancer progression in both haematological and solid cancers [26]. Further, pro-survival cellular processes such as autophagy, triggered chiefly by hypoxia, can be exploited by CSCs to sustain their survival [27]. In this review, we describe methods that have been used to identify CSCs and consider defining characteristics of CSCs in both solid and haematological cancers. Furthermore, we have sought evidence pertaining to the contribution of lncRNAs and autophagy in the maintenance of CSCs and how these regulatory factors and microenvironmental processes can affect outcomes of cancer therapy. We provide an appraisal of a computationally reinforced triangle inclusive of CSC properties, autophagy, and lncRNA and their associated networks with respect to hypoxia, epithelial-to-mesenchymal transition (EMT), and signalling pathways.

## 2. Methods for Detecting and Understanding the Characteristics of CSCs

If we concede that CSCs share qualities of tissue-specific stem cells, then it would be logical to test definitive markers and properties of these cells to identify CSCs. Indeed, one of the most widely used methods of detection and isolation of CSCs in cancers is by the detection of a cell surface expression profile reflective of the respective tissue-specific stem cell. Proteins such as CD44, CD90, and CD133 are regarded as common stem cell markers and are frequently employed to isolate CSCs in various cancer types (Table 1).

CSCs may also be characterised by their ability to efflux compounds, such as dyes, via ATP-binding cassette (ABC) transporters including MDR1/ABCB1 [45]. To that end, side population (SP) analysis was initially developed to enrich for haemopoietic stem cells in murine bone marrow and are detected by flow cytometry for quiescent cells that efflux Hoechst 33342 dye, with the non-stem cell population fluorescing post-excitation of the retained dye by UV light. This subpopulation within tumours may contain cells with stem-like characteristics [46]. This shows that SP analysis can enrich for a cell population with CSC properties although the SP may not exclusively contain CSCs. Another method to detect CSCs is the aldefluor assay which exploits the unique property of stem cells whereby high levels of aldehyde dehydrogenase 1 (ALDH1) activity are associated with stem-like properties. This enzyme is involved in the detoxification of aldehydes and retinoic acid synthesis, and may also contribute to the regulation of CSC self-renewal and differentiation, preventing damage from ROS and ultimately inhibiting apoptosis [47,48]. CSCs can also be detected by their ability to form tumours both in vitro and in vivo. Methods for addressing self-renewal and differentiation include sphere-forming and serial colony-formation assays showing not only anchorage-independent growth but also self-renewal capacity [49]. In addition, distinct characteristics of tissue-specific stem cells such as quiescence and dormancy have been applied to their identification, processes that can be regulated by changes to the epigenetic landscape, increased methylation of histones, and the formation of heterochromatin [50]. However, the caveat of simplistic 2D models is that they cannot effectively emulate the complexity of the tumour microenvironment (TME), and factors such as cell–cell and cell–matrix interactions, as well as a realistic distribution of oxygen, nutrients, or signalling proteins [51]. A popular assay that accounts for these factors is the multicellular tumour spheroid (MTS) method that bridges the gap between monolayer 2D cultures and in vivo models [52]. Finally, serial transplantation experiments whereby limiting dilutions of human tumour cells are propagated through immunodeficient mice can indicate the frequency of the CSC population. Furthermore, defined subpopulations of tumour cells can be propagated through immunodeficient mouse models (e.g., NOD/SCID) to assess them not only for self-renewal capabilities but also for their regenerative potential in turn producing a malignancy with the same cellular and molecular heterogeneity of the parent tumour [53]. CSCs can divide both symmetrically and asymmetrically in the xenografted immunodeficient mouse model, through which the new CSCs and daughter cells are produced [54]. However, one caveat is that the complex immune profiles of these models might underestimate the frequency of CSCs and lead to their misidentification. Overall, serial transplantation, cell surface protein expression, and in vitro assays, are collectively required to rigorously examine potential CSCs for stem-like properties.

## 3. Identification of Cancer Stem Cells in an Array of Cancers

In this section, we present evidence supporting the existence of CSCs and their characteristics across ten malignancies.

### 3.1. Acute Myeloid Leukaemia (AML)

Evidence supporting the existence of CSCs was first revealed in AML. Bonnet and Dick demonstrated heterogeneity in cancer cell populations in AML and furthermore that distinct cells within the leukaemic clone could give rise to AML in the NOD/SCID mouse xenograft model [17]. Subsequently, these cells were termed SCID leukaemia-initiating cells (SL-ICs) [17]. These SL-ICs were shown to exclusively express a CD34++CD38- surface profile similar to haemopoietic stem cells [17,28,29], although these data have since been disputed [30]. Overall, it is possible to conclude that despite the identification of subpopulations within the AML tumours with self-renewal capacity, universal markers of leukaemia stem cells (LSCs) remain inconclusive.

### 3.2. Oesophageal Cancer

Several proteins including CD44, MAML1, CD271, and CD90 have been proposed to identify CSCs in osophageal squamous cell cancer (ESCC) as has Hoechst 3342 dye efflux, in vitro colony formation assays, and xenotransplantation [55]. In particular, CD90+ oesophageal CSCs, isolated from primary tissue, were linked to the promotion of metastasis by deregulating ETS-1/MMP signalling pathways [56]. Other properties of oesophageal CSCs were reported as the ability to form spheres in vitro, possessing ALDH1 activity, and having distinct glycolysis and oxidative phosphorylation activities. These CSCs also displayed a dependence on signalling pathways such as the Hsp27-AKT-HK2 pathway [57]. In a follow-up study, B7H4 was identified as a novel marker of oesophageal CSCs that express SOX9 and OCT4 [31]. This group also showed that LETM1 may be a marker of CSCs in this cancer; expression of this protein positively correlated with that of the stemness marker OCT4 as well as cyclin D1, and was associated with hypoxia-inducible factor 1 alpha (HIF-1a) expression [32].

### 3.3. Colorectal Cancer

Colorectal cancer CSCs have been independently identified as cells in the bulk tumour expressing the specific markers CD133 and CD44, by sphere formation, ALDH1 activity, and the presence of SPs [58,59,60].

These cells expressed higher levels of ST6GALNAC1 had increased sphere-forming capacity and were resistant to therapy. From a mechanistic viewpoint, the group revealed that the role of ST6GALNAC1 in maintaining colorectal CSCs was modulated through the PI3-Kinase/AKT pathway [33].

### 3.4. Gastric Cancer

Gastric cancer CSCs were isolated from cell lines such as the SGC7901 cell line by pre-treatment with vincristine, thereby eliminating the non-CSC and enriching for the drug efflux pump-expressing, multidrug resistant CSCs [34]. Furthermore, the surviving non-CSC upregulated expression of OCT4, OCT4a, and SOX2, essentially dedifferentiating into stem-like cells [34]. Moreover, these cells expressed mesenchymal markers such as SNAIL, TWIST, and vimentin, and low levels of E-cadherin, while demonstrating an ability to form gastric crypt-like structures marked by expression of gastrointestinal genes such as *CDX2* and *SOX2* [34]. Finally, these CSCs displayed a marked capacity for tumourigenesis in vivo undergoing both symmetric and asymmetric division [34]. Other markers of CSCs in this cancer include CD44 and KLF4 [34,35].

### 3.5. Pancreatic Cancer

The identity of pancreatic adenocarcinoma CSCs was reported by Li and colleagues [36]. This group used xenotransplantation to identify a tumourigenic sub-population of cancer cells isolated from human primary pancreatic cancer tissue expressing CD44, CD24, and epithelial-specific antigen (ESA) [36]. This group reported that just 100 CD44+CD24+ESA+ cells were sufficient to faithfully capture the full characteristics of the primary human tumour in an orthotopic mouse xenograft model [36]. Furthermore, pancreatic CSCs expressing CD133 displayed tumourgenic properties and were resistant to chemotherapy (although these cells may represent persister cell populations rather than CSCs) [37].

### 3.6. Hepatocellular Carcinoma

Hepatocellular CSCs have been defined by the expression of cell surface proteins including CD13, CD24, CD44, CD90, CD133, and EpCAM [38]. Moreover, ALDH1 activity and Hoechst dye efflux are among other characteristics of these cells, while xenotransplantation has been used to rigorously test self-renewal capacity [38].

### 3.7. Lung Cancer

CSCs of lung adenocarcinoma were identified by ALDH1 expression levels whereby ALDH1-high populations isolated from primary tissue, expressing *SOX2* and *NANOG,* produced spheroids in culture [39]. Furthermore, CSCs isolated from NSCLC primary tissue showed unlimited growth capacity and self-renewal properties producing spheroids in vitro and tumours in vivo [40]. These CSCs were distinguished within tumourspheres as cells expressing NANOG, CD44, ITGA6, SNAI1, NOTCH3, and CDKN1A [40]. Moreover, CDKN1A, SNAI1, and ITGA6 expression levels were determined to be of prognostic value for patients and were utilised to determine a so-called “CSC score” [40].

### 3.8. Glioblastoma Multiforme

Cells expressing proteins including SOX2, OCT4, pSTAT3, KLF4, NANOG, and SALL4 were identified as CSCs in Glioblastoma multiforme (GBM) [42]. From a functional viewpoint, studies have shown the capacity of GBM CSCs (CD133+) isolated from primary tissue to form spheroids in culture [61]. Moreover, CD133, a protein also expressed by neural stem cells, further defines CSC in primary human GBM, as they generate tumours in NOD/SCID mice [62]. However, CD133 cells isolated from primary patient tissue have been shown to generate tumours on implantation into rat brains also giving rise to CD133+ cells [41]. Other proteins such as CD44 and KLRC3 have also been investigated as markers of a CSC population, although recent evidence suggests that CD44 is an unreliable marker; cells expressing low levels of CD44 isolated from patient samples, also display CSC characteristics [42,63,64].

### 3.9. Osteosarcoma

Human osteosarcoma primary tissue has been shown to contain SPs detected following Hoechst 33342 dye efflux. These SPs form colonies and self-renew both in vitro and in vivo in xenotransplantation experiments [46]. These SPs were further characterised as expressing ATP-binding cassette protein transporters and the stemness marker OCT4 [65]. Another protein associated with a CSC phenotype in osteosarcoma is SOX2 which was shown to maintain this cell population via modulation of the Hippo pathway [43]. In evidence, conditional knockout of *SOX2* in a mouse model of osteosarcoma resulted in a dramatic reduction in tumour development supporting the role of this stemness gene in maintaining tumour growth and providing evidence towards a population of cells with stem-like features in osteosarcoma [66].

### 3.10. Breast Cancer

The tumourigenic properties of breast cancer have been attributed to a population of CD44+CD24-/low cells (accounting for 11–35% of the bulk tumour), isolated from primary tumour tissue, that give rise to malignancies in xenograft models, following orthotopic transplantation into the mammary fat pad of NOD/SCID mice [13]. This cell phenotype has been further refined whereby CD44+CD24-/low cells expressing active ALDH1, enriched following doxorubicin treatment, were able to reproduce the bulk tumour in a mouse xenograft model [13,44].

## 4. Autophagy and CSCs

### 4.1. Autophagy: A Cellular Pro-Survival Process

Macroautophagy, hereafter called autophagy, is a catabolic cellular process by which cellular material is degraded. Derived from the Greek words auto, meaning “self” and phagein, “to eat”, autophagy is important in the maintenance of cellular homeostasis in eukaryotic systems [67]. It is one of two main routes by which proteins are degraded within the cell, the other being the ubiquitin proteasome system (UPS). The UPS deals with the rapid removal of short-lived proteins whereas autophagy concerns long-lived proteins, organelles, and non-protein targets (such as lipids and carbohydrates) [68]. The UPS consumes ATP in the degradation process while autophagy generates energy from the degradation of macromolecules [69]. The generation of energy through autophagy is important in the maintenance of cellular homeostasis.

Autophagy occurs constitutively at basal levels in nutrient-rich, healthy cells, and is stimulated when cells are exposed to stressors such as nutrient starvation, mitochondrial depolarisation, toxic protein aggregates, infection, and mechanical damage [70]. Depending on the nature and duration of the stress and the cell type, cells can display either protective or destructive autophagic responses. The initial response of a cell to a stressful stimulus is often the promotion of pro-survival actions. If these are unsuccessful, destructive pathways such as apoptosis, necrosis, and pyroptosis are activated, which eliminate these damaged cells [71]. Autophagy is largely regarded as a pro-survival process, although it can lead to cellular demise.

The process of autophagy involves the formation of a double membraned vesicle called an autophagosome, which encapsulates cellular components earmarked for degradation (Figure 2A) [72,73,74]. This process is modulated by a family of proteins termed “autophagy-related” (Atg) proteins. Autophagy induction takes place when signalling pathways trigger the formation of a membrane invagination described as an omegasome, mainly observed on the surface of the ER. However, other membranes such as the plasma membrane have been shown to act as sites of initiation [75]. This structure enlarges into a phagophore (either independently of, or associated with, the ER) and targets cellular material in close proximity for digestion. The phagophore elongates and eventually forms the autophagosome [76]. The mature autophagosome then becomes acidified following fusion with a lysosome in a Ca^2+^ -dependent manner to produce autophagosome [77]. This fusion unites the degradative acid hydrolases with the captured cell material. Following the breakdown of the cellular material, nutrients such as amino acids and fatty acids are exported back into the cytoplasm by lysosomal permeases [78]. As with other intracellular trafficking events, autophagosomes move through cytoskeleton-dependent action. This movement involves microtubule-associated proteins such as LC3 and actin microfilaments [79].

### 4.2. CSCs Rely on Autophagy for Stemness, Invasion, Migration, and Chemo-Resistance

Over the past few years, autophagy has been shown to be vital for the maintenance of stemness in both normal tissue stem cells and CSCs [80]. CSCs rely on autophagy for various roles; they use it as a means to survive, adapt to the tumour microenvironment, for migration and invasion as well as a route to escape radiation treatment and chemotherapy (Figure 2B). For many cancer types, including those of the pancreas, breast, bladder, colorectum, and brain (glioblastoma), CSCs are dependent on autophagy for maintaining their stemness. The basal level of autophagy/mitophagy is frequently higher in CSCs compared to that of normal tissue-specific stem cells [81]. Why stem cells are more reliant on autophagy is a current area of interest for many scientific groups. For instance, the interaction between transcription factor (TF) families in regulating CSC properties and autophagic processes has been reported. Forkhead box O (FOXO) TFs regulate autophagic proteins such as Beclin-1, LC3, ULK1, Atg5, Atg8, GABARAPL1, Atg12, Atg14, and BNIP3, while also regulating CSC properties [82]. FOXO3 is required by leukaemia-initiating cells for maintaining stemness, while in ovarian cancer stem cells (OCSCs), FOXO TFs are regulated by autophagic processes. Further, the lentiviral knockdown of *ATG5* led to a decrease in OCSC chemoresistance and the ability to self-renew [83,84].

Further, autophagy stimulates the expression of both stem cell markers such as CD44 and mesenchymal markers such as vimentin [85]. Cufi and colleagues were the first to demonstrate a link between autophagy and the maintenance of tumours, whereby autophagy promoted higher expression levels of CD44 and vimentin in BCSCs (CD44+CD24-/low cells) isolated from the JIMT-1 epithelial breast cancer cell line. Furthermore, inhibition of autophagy preferentially potentiated epithelial-like characteristics, marked by CD44+CD24+ surface marker expression, over the mesenchymal phenotype marked by low expression of CD24. By inhibiting autophagy using Chloroquine or by downregulating *ATG12*, the migration and invasiveness of these CSCs was impaired, as was vimentin expression [85].

Additionally, the autophagy regulators SQSTM1 and DRAM1 are highly expressed in GBM CSCs, and their expression correlates with that of mesenchymal factors such as c-MET [86].

There is increasing evidence to suggest that signalling pathways leading to autophagy and epithelial-to-mesenchymal transition (EMT) are linked. EMT is a vital event during embryonic development and is a critical feature linked to the ability of CSCs to migrate [13,87,88]. Indeed, inhibiting autophagy in some solid tumours such as GBM and breast cancer led to impairment of CSC migration. In addition, Helicobacter pylori-induced autophagy is implicated in the emergence of gastric CSCs. For instance, treatment of CSCs, characterised by CD44+ expression and tumoursphere formation capacity with autophagy inhibitors, reduced the appearance of a mesenchymal phenotype and tumoursphere formation [89].

Recently, studies of BCSCs showed that autophagy plays a role in mediating tumour dormancy. BCSCs can lie dormant for many years before recurring in metastatic lesions and resort to an increased autophagic flux to survive harsh tumour environments and to maintain their phenotype, in particular by resisting chemotherapeutics and hypoxia [90]. Furthermore, the autophagy-associated proteins Atg5, Atg12, and LC3-B are overexpressed in dormant stem cell-like breast cancer cells, a phenotype that can be reversed by 3-methyladenine (3-MA), an inhibitor of autophagy. Moreover, the JNK/SAPK signalling pathway is upregulated in these dormant stem cell-like breast cancer cells and is responsible for increasing autophagy amongst the population [91]. Similarly, the autophagy protein Beclin-1 is expressed at higher levels in ALDH1+ BCSCs derived from mammospheres, compared to tumour cells in the bulk population [92]. It therefore follows that inhibition of autophagy by knockdown of *ATG7* or *Beclin-1* reduces the secretion of IL-6, which has a crucial role in the maintenance of BCSCs [93].

A link between CSCs, autophagy, and drug resistance has been observed in numerous human cancers, including leukaemia, breast, pancreatic, urinary bladder, colon, and brain cancers [94], in which the preclinical and clinical use of autophagy inhibitors in combination with targeted therapies has been the main focus [95]. Indeed, combinations of cytotoxic drugs and autophagy inhibitors enhanced CSC sensitivity. For instance, in GBM CSCs, the EGFR inhibitor, TMZ, combined with Chloroquine inhibited proliferation [96]. In gastric CSCs, Chloroquine and 5-fluorouracil (5-FU) inhibited Notch signalling and reduced cell viability [97]. In bladder CSCs, siRNA-mediated and pharmacological inhibition of autophagy restored the anti-proliferative effects of the chemotherapeutic agents gemcitabine, mitomycin, and cisplatin [94].

Interestingly, studies have indicated that autophagy induction is involved in drug-induced cytotoxicity. For instance, Resveratrol, a natural polyphenolic compound, which triggers autophagy by blocking Wnt signalling inhibits breast CSC growth and survival [98,99]. In comparison with Chloroquine, newer lysosomal inhibitors such as Lys05, an analogue of Chloroquine, are more selective and potent. Indeed, in leukaemia, Lys05-mediated autophagy inhibition decreased the numbers of LSCs in vitro and in vivo [100,101]

### 4.3. Autophagy and the Hypoxic Microenvironment

In the tumour microenvironment, hypoxia is a known inducer of autophagy, mediated by HIF-1α. HIF-1α controls the expression of genes involved in initiating and maintaining CSCs such as OCT4, SOX2, KLF4, MYC, NANOG, ALDH1A1, and NOTCH [102,103,104]. In addition, hypoxic stress induces EMT TFs such as SNAIL, TWIST, and ZEB2 which are direct targets of HIF-1α [105,106,107]. Other HIF-1α target genes including BNIP3/BNIP3L mediate autophagy induction under hypoxic conditions, leading to cell survival [108], while hypoxia-induced NANOG can bind the promoter of BNIP3L, revealing a regulatory loop [109].

CSCs are particularly reliant on autophagy for survival in hypoxic states. Of note, liver CD133+ CSCs, as well as pancreatic CSCs, critically rely on hypoxia-induced autophagy for their survival [110]. Interestingly, Zhu and colleagues showed that intermittent hypoxia reprogrammes non-stem pancreatic cancer cells into pancreatic CSCs, with increased autophagic flux and HIF-1α levels [110]. Additionally, gene expression analysis showed autophagy to be one of the major pathways induced by hypoxia in colon CSCs; PRKCA/PKCα was shown to be involved in hypoxia-induced autophagy-mediated CSC self-renewal whereby knockdown of *ATG5* significantly reduced in vivo tumour formation [111]. Autophagy was also found to be upregulated in multiple human AML cell lines following exposure to hypoxia. Inhibition of the late stage of autophagy with either Chloroquine or Bafilomycin A1 treatment in leukaemia stem cells (LSCs) allowed for these cells to overcome hypoxia-induced resistance to cytarabine (AraC) [112].

With respect to the signalling pathways involved in these processes, the PI3K/Akt/mTOR was shown to be inhibited by hypoxia [113], while mTOR was found to interact with and regulate HIF-1α [114,115]. In lung cancer cells, the drug Gigantol targets CSCs by inhibiting PI3K/AKT/mTOR and JAK/STAT pathways [116]. Other targets of HIF-1α, *KLF5*, were shown to be upregulated in response to hypoxic stress [117], while siRNA-mediated *KLF5* knockdown, inhibited hypoxia-induced cell survival and promoted apoptosis through the inactivation of the PI3K/Akt/mTOR pathway [118].

### 4.4. CSCs Are Dependent on Mitophagy for Their Metabolic Reprogramming

To maintain their proliferative needs, cancer cells rely on a constant nutritional supply by using various strategies such as redox signalling, a high glycolytic flux, and autophagy. Unlike most cancer cells, which mostly depend on aerobic glycolysis, CSCs show a mixed phenotype where both CSCs undergoing OXPHOS metabolism (e.g., breast, pancreatic, and lung CSCs) or glycolytic metabolism (e.g., glioma stem cells (GSC)) have been demonstrated in different cancer models [81].

In addition to autophagy, mitophagy has been shown to be a key mechanism in the homeostasis and invasion of CSCs [119]. Since mitochondria are essential for regulating cellular energy homeostasis, mitophagy has been the subject of extensive research and has recently been proposed to play critical roles in many diseases [120]. As the name suggests, mitophagy is the selective degradation of mitochondria by autophagic processes. In addition to maintaining the stability and integrity of mitochondrial function and structure, mitophagy is important for the maintenance of mitochondrial number [121] and the elimination of mitochondria during the development of specialised cells such as reticulocytes [122]. Mounting evidence shows that CSCs critically depend on mitochondrial function for their survival, migration, as well as resistance to toxic agents [123,124,125]. Moreover, CSCs rely on mitophagy in order to keep ROS levels under control and consequently in the prevention of DNA damage and the induction of apoptosis [126].

Mitochondrial fission, mediated by dynamin-related protein 1 (DRP1) has been shown to facilitate mitophagy by dividing mitochondria into fragments ready for autophagosomal engulfment [127,128]. Glioblastoma CSCs, show activation of DRP1 which correlates with poor survival in GBM [129]. In addition, mitophagy has been shown to regulate hepatic CSCs by inhibiting the tumour suppressor p53 and promoting the expression of NANOG [130]. Apart from playing a role in the survival of CSCs, mitophagy has also been implicated in chemoresistance mediated by metabolic reprogramming, whereby mitophagy contributes to doxorubicin resistance of colorectal CSCs [131]. Indeed, a combination of mitophagy inhibitors with anti-tumour drugs increases CSC death [132]. For instance, the use of the classic chemotherapeutic drugs cisplatin, doxorubicin, and vincristine, in combination with the mitophagy inhibitor liensinine increased the sensitivity of breast cancer stem cells to treatment [133].

## 5. Long Noncoding RNAs and Cancer Stem Cells

### 5.1. Long Noncoding RNAs in Health and Disease: A Central Role in Cancer

Long noncoding RNAs (lncRNAs) are transcripts longer than 200 nucleotides that lack protein-coding potential. Following their discovery, their functional role has been questioned [134,135]. LncRNAs are nowadays some of the most investigated molecules in health and disease and have shown key potential as cancer biomarkers and therapeutic targets. In fact, lncRNAs can be expressed in a tissue- and disease-specific manner showing aberrant up- or down-regulation in specific cell-types and different malignancies [136,137,138]. Despite numerous studies having characterised lncRNAs in cancer, their expression patterns and roles are still largely uncharacterised. For the most part, this is due to the abundance of lncRNAs (more than 50,000 estimated encoded by the human genome) to be studied and their ability to simultaneously act in different cellular pathways with various mechanisms of action [139]. Among the most well characterised functions, lncRNAs can act in epigenetic regulation, transcriptional and post-transcriptional regulation, miRNA synthesis and function, and protein scaffolding, thereby regulating fundamental cellular processes such as chromatin organisation, cell proliferation and survival, cancer growth, metastasis, and drug resistance [140,141,142,143]. CSCs play key roles in tumour-associated processes but the mechanisms regulating and maintaining CSC characteristics are largely unknown [140]. Notably, numerous lncRNAs modulate tumour initiation, progression, metastasis, and drug resistance, via altering the expression of self-renewal factors and ultimately the functions of CSCs [144].

### 5.2. LncRNAs in CSC Maintenance and Migration

#### 5.2.1. Epigenetic Regulators

LncRNAs can be involved in cancer through epigenetic regulation in CSCs since they can promote epigenetic factor and co-factor expression, stability, and function. The *X-chromosome inactivating specific transcript* (*XIST*) gene is localised in the X-chromosome inactivation centre (Xic), encodes for the *XIST* lncRNA, and is associated with tumour regulation in various malignancies. In ovarian cancer, *XIST* promotes the stability of lysine (K)-specific methyltransferase 2C (KMT2C) mRNA, thereby indirectly promoting histone H3 methylation at lysine 4 and hence decreasing CSC proliferation [145]. *XIST* can also alter the phenotype of CSCs by increasing paclitaxel sensitivity in patients; *miR-93-5p* can reverse this pathway, demonstrating multifaceted interactions between epigenetics and lncRNAs [145].

Histone methylation is a known epigenetic mechanism that induces transcriptional repression via chromatin modification. In human liver CSCs, transcription of the lncRNA *CUDR* was stimulated by TLR4, which promoted their proliferation in vitro and growth in vivo, via control of histone methylation and telomere elongation [146]. TLR4 action also prompted a stable interaction between *CUDR,* SUV39 h2, and histone 3, which led to epigenetic repression in liver CSCs [146]. This function, mediated by this lncRNA, controlled downstream pathways, including telomere length, playing a fundamental role in controlling CSC survival [146] (Figure 3).

*LncHDAC2* is another lncRNA upregulated in liver CSCs where it promotes stem cell self-renewal by activating hedgehog signalling via acting as an epigenetic co-effector [137]. *LncHDAC2* recruits the Nucleosome Remodelling and Deacetylase (NuRD) complex to the promoter of *PTCH1*, a component of the hedgehog signalling pathway. Since liver CSCs harbour resistance to common therapies, inhibition of their self-renewal capability via *lncHDAC2* targeting could suppress liver tumorigenesis [137]. Other lncRNAs also regulate epigenetic modifications in liver CSCs such as *HOTAIR,* which acts by similar mechanisms in other CSCs, such as those found in prostate cancer, that may express c-KIT surface marker [147,148]. These studies highlighted the promiscuity of lncRNAs that can regulate the same specific pathways in one or more cancer types, making their study of important translational potential for different malignancies.

#### 5.2.2. miRNA Synthesis and Function

Most lncRNAs have been characterised as competing endogenous RNAs (ceRNAs) since they can compete for miRNA binding, thereby sequestering these small ncRNAs from binding the 3′UTR of target mRNAs. In this way, lncRNAs can indirectly promote mRNA stability and protein translation. *H19* was characterised in several cancers as being oncogenic and a promoter of metastasis but its mechanisms of action were only partially elucidated. In pancreatic ductal adenocarcinoma, *H19* regulates metastasis in vitro and in vivo by increasing the self-renewal potential of CSCs, sphere-formation, and the ability to invade into surrounding tissues [149]. *H19* also promotes sphere formation by pluripotent CSCs produced from human mammary epithelial MCF-10A cells and its expression is associated with reduced disease-free and overall survival in breast cancer patients [150]. *H19* is a precursor of *miR-675* and both of these ncRNAs promote breast cancer metastasis via induction of EMT and features of breast CSCs, both in vitro and in vivo [141].

Another *lncRNA, SOX2OT* promotes bladder cancer cell stemness, thereby inducing tumour growth and metastases in vivo with its expression linked to unfavourable clinical attributes, such as high histological grade, advanced TNM stage, and a poor prognosis [151]. *SOX2OT* is a ceRNA that sequesters *miR-200c* and upregulates SOX2 expression, a fundamental transcription factor for stem cell self-renewal. SOX2 overexpression in turn reverses *SOX2OT* silencing-induced inhibition of the bladder CSC stemness phenotype [151] (Figure 3).

Crosstalk between ncRNAs can also occur following upstream targeting by miRNAs of lncRNAs, such as *CCAT2* and *LUCAT1*. *CCAT2* is an lncRNA aberrantly expressed in triple-negative breast cancer (TNBC), which is known to be highly aggressive [136]. *CCAT2* levels are higher in breast CSCs while promoting the expression of CSC-associated proteins such as OCT4, NANOG, and KLF4, and inducing mammosphere formation [136]. *CCAT2* upregulates the expression of the OCT4 pseudogene, *OCT4*-*PG1,* which in turn is targeted by *miR-205*. *miR-205* is also known to target *NOTCH2*, another gene upregulated by *CCAT2* in TNBC [136]. Likewise, *LUCAT1* is an lncRNA associated with clinical features of breast cancer patients, such as tumour size, lymph node metastasis and TNM staging, and a poor prognosis [152]. In vitro, *LUCAT1* promotes breast cancer cell proliferation and is aberrantly upregulated in breast CSCs where it promotes self-renewal by regulating the Wnt/β-catenin pathway [152]. *LUCAT1* can be targeted by *miR-5582* which inhibits its expression. Hence, *LUCAT1* represents a putative prognostic biomarker and a novel target for treatment strategies in clinical practice [152].

#### 5.2.3. Transcriptional Regulators

LncRNAs play multifaceted roles in transcriptional regulation, from the stabilisation of transcription factors to modulation of gene expression and splicing. HIF transcription factors play a central role in promoting hypoxia and participating in other oncogenic pathways [143,153]. In renal cell carcinoma, HIF transcription factors affect the expression of the stem cell transcription factor OCT4, which correlated with advanced tumour stage and poor overall survival of renal cell carcinoma patients [143]. *PSOR1C3* is an lncRNA upstream of OCT4, containing an HIF-responsive long terminal repeat (LTR) element in its promoter. Therefore, *PSOR1C3* expression is regulated by HIF transcription factors and can in turn affect transcription of OCT4 variants, due to a transcription factor-lncRNA-transcription factor-mediated positive regulation mechanism [143] (Figure 3).

Although most lncRNAs have been shown to promote oncogenic factors and processes, some lncRNAs can also hinder cancer and CSC features. An example is the lncRNA *lnc-DILC* [154]. Microarray and RT-qPCR validation, conducted on liver CSCs, revealed downregulation of *lnc-DILC; Lnc-DILC* inhibited the expansion of liver CSCs via inhibition of IL-6/STAT3 signalling, by directly binding the *IL-6* promoter [154]. *Lnc-DILC* was also downregulated in hepatocellular carcinoma patient tumours and correlated with patient prognosis [154].

#### 5.2.4. LncRNA-Mediated Regulation of CSCs Modulates Drug Resistance

Several studies have implicated lncRNAs in cancer drug resistance and recent findings showed that lncRNAs can modulate treatment-resistant phenotypes via regulation of CSC biology and function [145,155,156].

In evidence, for pancreatic cancer, the lncRNA *RP11-567G11.1* is upregulated in poorly differentiated tissues and promotes proliferation and cell cycle progression, induces apoptosis and a stem cell-like phenotype by triggering factors downstream of the Notch signalling pathway [155]. Depletion of *RP11-567G11.1* increased gemcitabine response suggesting its potential value as a therapeutic target in drug-resistant tumours [155]. Furthermore, microarray analysis of cholangiocarcinoma and adjacent healthy tissues revealed differential expression of the lncRNA *lnc-PKD2-2-3*, which was confirmed in 60 paired samples by reverse transcription-quantitative polymerase chain reaction (RT-qPCR) [157]. Modulation of *lnc-PKD2-2-3* expression via lentiviral overexpression or hairpin RNA silencing revealed that this lncRNA induced the expression of CSC identifying proteins such as CD44, CD133, and OCT4, thereby inducing sphere formation and drug resistance to 5-fluorouracil (5-FU) [157]. In this study, *lnc-PKD2-2-3* was confirmed to promote oncogenic and stem-like characteristics such as poor tumour differentiation, advanced TNM stage, increased carcinoembryonic antigen expression, and was associated with poor overall survival for cholangiocarcinoma patients. Indeed, *lnc-PKD2-2-3* was upregulated in cholangiocarcinoma stem-like cells [157]. This lncRNA could be a future therapeutic target and biomarker of CSC prevalence in cholangiocarcinoma and in other tumours where it promotes stemness [157].

In pancreatic cancer, the lncRNA *MALAT-1* exerts oncogenic functions via promotion of EMT and stimulation of CSC-associated protein expression suggesting a role for this lncRNA in stimulating stem-like phenotypes [140]. *MALAT-1* promoted spheroid formation and resistance to gemcitabine in pancreatic cancer cells as well as tumorigenicity in vivo by stimulating the expression of self-renewal-associated stem cell transcription factors and other proteins, including OCT4, NANOG, SOX2, BMI1, β-catenin, and c-Myc [140]. The roles of lncRNA in cancer aggressiveness such as drug resistance is of fundamental importance to finding novel therapeutic approaches to overcome resistance and increase patient survival.

### 5.3. LncRNAs as Cancer and CSC-Specific Therapeutic Targets and Biomarkers

Since lncRNAs do not encode for proteins, classical pharmacological treatments targeting protein structure or activity are not effective. Nevertheless, this paves the way for the use of novel approaches that could be tested in clinical trials to modulate lncRNA expression, with promising advantages for precision medicine. Many lncRNAs are upregulated as components of specific oncogenic pathways in CSCs. Therefore, their inhibition could be exploited for therapy either as a single target or in combination with other treatments to improve patient response to therapy and survival, potentially with reduced side-effects. At present, therapeutic targeting of lncRNAs is largely being explored at a preclinical stage, although some successful methods of lncRNA inhibition are currently undergoing studies in clinical trials, such as the use of antisense oligonucleotides (ASOs) [158,159]. ASOs are single-stranded DNA polymers (13-200nt), which are readily internalised by cells in which they bind to their target RNA, thereby inducing DNA/RNA complex degradation, mediated by RNase-H [160]. RNase-H is an enzyme expressed both in the nucleus and cytoplasm [161], thereby leading ASOs to target both nuclear and cytoplasmic lncRNAs. Features of ASOs have been improved in recent studies via chemical modification. In this way, a new generation of antisense molecules has been created, such as locked nucleic acids (LNAs), Phosphorothioate (PS) ASOs, and morpholino oligomers. LNAs are single-stranded oligonucleotides with increased stability and strength of hybridisation due to a stretch of DNA flanked by LNA nucleotides which confer specific complementarity and RNase H-mediated degradation of the target sequence [162]. PS ASOs have increased molecular stability, due to the substitution of one oxygen of the phosphodiester bond between two ribose molecules with a sulphur group that creates a phosphorothioate bond [163], thereby increasing molecular resistance to nuclease digestion and stronger serum protein binding. This provides PS ASOs with increased stability in the circulation promoting tissue and cellular uptake [163]. Similar characteristics can be obtained by other chemical changes to obtain morpholinos, characterised by the replacement of deoxyribose with a six-membered morpholine ring, and of the charged phosphodiester inter-nucleoside linkage with phosphorodiamidate linkages. Therefore, a non-ionic structure is created of 25bp length, able to sterically block ribosomal assembly, affect RNA splicing and bind to complementary RNAs with increased affinity and stability, upon serum and plasma nuclease and protease activity [164,165]. Morpholinos have been successfully used for targeting c-Myc in lung and prostate cancer preclinical models with a phase I trial underway for prostate cancer [164,165]. The effect of antisense molecules has also been tested for lncRNAs in cancer and specifically CSCs. *NRAD1* is an lncRNA upregulated in TNBC CSCs and is associated with a poor patient prognosis [166]. Antisense oligonucleotides targeting *NRAD1* reduced survival and TNBC tumour growth as well as affecting CSC characteristics by regulating nuclear functions such as enriching chromatin interactions [166]. Other methods have been considered for targeting lncRNAs with a therapeutic aim, such as the transient use of siRNAs or stable transduction with lentiviral vectors or CRISPR-Cas9 genome editing, but their use in the clinic is still a way off [167,168,169].

Closer to clinical investigation and approval is the use of lncRNAs as cancer and CSC biomarkers in patients. As mentioned previously, lncRNAs can be aberrantly upregulated in specific cancer cells, including CSCs [136,137,138]. Due to their specific expression, they are optimal candidates for prognostic and diagnostic biomarkers. Clinical trials are currently recruiting patients for lncRNA-based cancer diagnosis (https://clinicaltrials.gov/ct2/show/NCT03830619 accessed on 5 February 2019 https://clinicaltrials.gov/ct2/show/NCT04269746 accessed on 8 July 2020) and the lncRNA *PCA3* was also clinically approved for cancer diagnosis [170]. Notably, lncRNAs were detected in biological fluids, suggesting that they could be used as non-invasive biomarkers for different malignancies [171,172]. *H19* stimulates the expression of stemness markers such as CD133, NANOG, OCT4, and SOX2 inducing the clone-forming ability of glioma and breast CSCs [144].

*HOTTIP* is highly upregulated in pancreatic CSCs and promotes stemness features, such as the ability to form spheroids, tumours, and expression of stemness-related transcription factors (LIN28, NANOG, OCT4, and SOX2) and other proteins (ALDH1, CD44, and CD133) via binding WDR5 and so stimulating HOXA9, activating the Wnt/β-catenin pathway. The role of *HOTTIP* was confirmed in vitro using human pancreatic cancer cell lines (i.e., PANC-1 and SW1990) in promoting sphere formation, and in vivo by growth of engrafted sphere cells [173].

Since lncRNAs can promote the expression of proteins associated with the CSC phenotype, their detection could aid in the identification of CSCs in patient-derived tumours by simple non-invasive diagnostic tests. In evidence, several studies have found lncRNAs secreted in tumour-associated exosomes secreted into the peripheral circulation [171,174,175]. In this regard, a recent study showed that in hepatocellular carcinoma, *H19* was upregulated in CD90+ CSC-like cells and was released in exosomes [172]. These cells were able to modulate the surrounding microenvironment thereby promoting angiogenesis and cell adhesion. Moreover, modulation of *H19* expression showed that this lncRNA affected the exosome-mediated mechanism of angiogenesis [172]. Exosomes can facilitate the transport of lncRNAs away from the tumour microenvironment, making their non-invasive detection a real opportunity for the clinical diagnosis of cancer and the identification of CSCs.

## 6. The Interrogation of Gene–Disease Networks for CSC-Associated Genes

In this section, we have endeavoured to draw links between the CSC factors and markers of stemness with prominent genes involved in CSC-related autophagy processes, interaction with the microenvironment, and CSC-associated lncRNAs. To that end, we conducted network analyses using multiple platforms on genes discussed in this review, inclusive of *CD34*, *CD38*, *B7H4*, *LETM1*, *CD90*, *ST6GALNAC1*, *CD44*, *SOX2*, *KLF4*, *OCT4*, *CD133*, *CD24*, *ESA*, *CD13*, *EpCAM*, *NANOG*, *ITGA6*, *SNAI1*, *NOTCH3*, *CDKN1A*, *c-KIT*, *STAT3*, *SALL4*, *ALDH1*, *vimentin*, *ATG5*, *ATG12*, *LC3-B*, *Beclin-1*, *ATG7*, *SQSTM1*, *DRAM1*, *c-MET*, *HIF-1α*, *PRKCA*, *DRP1*, *XIST*, *CUDR*, *LncHDAC2*, *HOTAIR*, *H19*, *miR-675*, *SOX2OT*, *CCAT2*, *PSOR1C3*, *miR-205*, *NOTCH2*, *miR-5582*, *lnc-DILC*, *RP11-567G11.1*, *lnc-PKD2-2-3*, *MALAT-1*, *BMI1*, *c-Myc*, *HOTTIP*, *KMT2C*, *miR-93-5p*, *TLR4*, *SUV39 h2*, *PTCH1*, *mir-200c*, *LUCAT1*, *OCT4-PG1*, *β-catenin*, *NRAD1*, *LIN28*, *WDR5*, *HOXA9*, and *Wnt.*

We used the Reactome database in Cytoscape to interrogate the functional interactions between these genes, under two assumptions; A) genes that were not included in this study could be added to the network, or B) only genes included in this study could be included in the network (Figure 4) [176,177,178]. These gene products may be involved in several interconnected biochemical pathways that may be activated in different contexts. For instance, there is evidence that SOX2 may activate *NANOG*, or that PRKCA may activate *ITGA6*. Collectively, these data demonstrate the functional association between these genes.

We extended our network analysis to understanding the functional and physical protein–protein interactions using STRING v11.04 (Figure 5A) [179]. For the submitted list, the results indicate a greater than expected interaction for a network of this magnitude, suggesting a highly interactive group of proteins (51 nodes, 70 interactions, 15 expected interactions, *p*-Value < 10^−16^). For instance, SNAI1 strongly interacts with SALL4, while the interaction between STAT3 and Beclin-1 is weaker.

In addition, we sought to interrogate disease–gene interactions of the genes mentioned above, utilising DisGeNet database using clusterProfiler package [180,181]. This analysis revealed the most highly enriched diseases, represented by gold nodes, that were associated with these genes. These include gastric adenocarcinoma and urothelial carcinoma (Figure 5B) (Bonferroni correction *p*-Value < 0.0005). Finally, we found transcription factor targets and putative miRNA target predictions by Genemenia using MSigDB8 databases, with targets displayed in red [182,183]. For instance, *mir449* is a target of KLF4 and MET (Figure 6).

Using these tools, we have sought to drive the message that the stemness genes, genes involved in the maintenance of stemness, genes associated with autophagy-related processes and lncRNAs, are strongly interlinked and also show a strong association with cancers. This interlinkage is multifaceted, ranging from co-expression to physical binding events to interactions along biochemical pathways mediated by additional proteins. Ultimately, these data show that a complex interplay between signalling pathways drives tumour growth in heterogeneous cell populations and therefore that any therapeutic approach must take into account all of these factors to be successful.

## 7. The Triangle of CSCs, Autophagy, and lncRNA

In light of the evidence presented in Section 2, Section 3, Section 4, Section 5 and Section 6, we aim to further substantiate the links between the triangle of CSC phenotype/potential, autophagy, and lncRNA and their networks. We learned that autophagy, chiefly triggered by hypoxia, is employed by CSCs as a means to adapt to harsh TME conditions and can (through HIFs) maintain stemness factors. Hypoxia-triggered HIFs, then directly or through their extensive targets, stimulate the maintenance of stemness markers such as OCT4, SOX2, KLF, NOTCH, and NANOG [27,184]. HIF target genes in turn could induce autophagy under hypoxic conditions, demarcating a feedback loop. On the other hand, the role of lncRNAs in CSC phenotype establishment such as their contribution to CSC self-renewal has been reported. Here we reviewed the regulation of stemness markers such as NANOG and OCT4 by *H19*, *HOTTIP,* and *CCAT2* [136,144,173]. In addition, *MALAT-1, H19,* and *CUDR* can maintain the length of telomeres, therefore contribute to CSC subpopulation maintenance [25]. These observations highlight the contribution of two key mechanisms to the maintenance of stemness markers such as OCT4 and NANOG, by hypoxia-driven HIF TFs (and through that autophagy), or by *H19, HOTTIP,* and *CCAT2* lncRNAs. Indeed, cooperation between lncRNAs and HIFs have been documented. For instance, P53-HIF-H19/IGF2 signalling pathway has been described in glioma [185], while it has been reported that HIFs can directly bind to the promoter of *PSOR1C3* to promote stemness factors [143]. Collectively these observations form a triangle inclusive of hypoxia-driven HIFs, lncRNA, and autophagy, that through interlinked mechanisms, can maintain CSC stemness factors. In Figure 7, we have sought to summarise the triangle of the interactions between CSC phenotype/potential, autophagy, and lncRNA with respect to stemness markers.

Another interesting aspect may be the convergence of the members of this triangle, on the EMT axis. EMT is triggered by hypoxia and it can trigger TFs such as SNAI1 and TWIST1. Further, key mediators of signalling pathways or stemness factors inclusive of TGF-B, STAT3, Notch, and NANOG, are triggered by EMT that can in turn contribute to the maintenance of the CSCs [27,184,186]. Activated STAT3 can contribute to EMT and can bind to the promoter of *TWIST1*. Further, STAT3 in cooperation with HIF-1α can increase the expression of CD133, a marker of CSCs [187]. Moreover, Atg4, Beclin-1, and, P62 can influence EMT and contribute to CSC maintenance in breast cancer [188], while in pancreatic cancer, lncRNA *MALAT-1*, can promote EMT and CSC-associated proteins [140]. Collectively, we can conclude that lncRNA and autophagy, and hypoxia-induced HIF-1α can influence EMT through interconnected mechanisms and signalling pathways, leading to the maintenance of CSC properties (Figure 8).

Finally, we hypothesise that CSCs, autophagy, and lncRNA may collaborate on the axis of inflammation. In a study utilising the co-culture of MCF-7 breast cancer cell line and immortalised foreskin fibroblasts, inflammatory mediators such as IL-6 and IL-8 were upregulated in the extracellular matrix, while autophagy was induced due to oxidative stress [189]. We also learned that in BCSCs derived from mammospheres, the inhibition of autophagy led to the reduction in IL-6 secretion affecting the maintenance of these CSC [92,93]. Hence, we can hypothesise that through crosstalk between CSCs and cancer-associated fibroblasts, or perhaps other mechanisms, cytokines such as IL-6 can support CSC maintenance. In addition, an lncRNA, *lnc-DILC*, that through binding to the promoter of IL-6 reduced the expansion of liver CSCs [154], suggesting potential negative feedback loops. Through these examples, we have endeavoured to further substantiate the notion of functional links between lncRNA, autophagy, and CSCs through interaction with the microenvironment.

## 8. Discussion and Conclusions

If we accept that a sub-population of cancer-initiating cells or CSCs are endowed with capacities to initiate or propagate tumours, respectively, and to self-renew, while persister cells resist treatment, then directing our collective efforts towards developing and fine-tuning treatment strategies that specifically target and eradicate these cells is paramount [33]. Irrespective of theoretical and practical definitions of these cells with stem-like characteristics and related disputes, treatment response, metastasis, evasion of immunosurveillance, and the resulting impact on patient prognosis and well-being are important topics that the cancer community faces. Failed cancer therapies, in many cases, are due to the inability of the treatment to eliminate the CSCs, leading to inevitable relapse. For instance, many cancer therapies are aimed at targeting highly proliferative cells of the cancer population and thereby may not effectively target quiescent CSCs [190]. One good example of this is the treatment of pancreatic cancer cells in vitro and in vivo with gemcitabine. This effort led to apoptosis only in the non-CSC population of the tumour in vitro and led to the enrichment of a CD133+ CSC fraction in vivo in the mouse xenograft model [37]. This and numerous other studies indicate large gaps in our understanding of CSCs, their associated cellular processes, interaction with the microenvironment, and regulatory mechanisms that govern their maintenance. CSCs in their TME are affected by a myriad of intrinsic and extrinsic cues. Examples of extrinsic factors include hypoxia, stress stimuli, nutrient supply, or variable levels of growth factors. Concomitantly, these environmental factors can influence the CSCs to activate stress response pathways to circumvent cell cycle arrest or death. For instance, CSCs in response to hypoxia can instigate an HIF-1α -mediated angiogenesis processes to promote survival [27]. Moreover, dormancy, maintenance, and survival in the harsh tumour microenvironment for CSCs is dependent on autophagic processes [90,92,93]. CSCs in the liver and pancreas rely on hypoxia-induced autophagy for survival and reprogramming [110], hence, as discussed, inhibiting autophagy as a pro-survival mechanism of CSCs, in the form of combination therapy may increase therapy success. For instance, in a study conducted by Jang and colleagues, the apoptotic rate and autophagy induced by a bromodomain and extraterminal domain (BET) inhibitor, JQ1, in CD34+ CD38- leukaemic blasts (LSCs) in AML were investigated, although as mentioned in Section 3, the universal markers of CSCs in this cancer remain inconclusive [30,191]. The authors reported that JQ1 increased cell death in JQ1-sensitive LSC blasts [192]. However, in JQ1-resistant AML LSCs, autophagy mediators such as Beclin-1 were upregulated, while lipidation of LC3 was increased, whereas only minor levels of apoptosis were detected. Interestingly, the inhibition of autophagy led to increased apoptosis in the JQ1-resistant LSCs, revealing the potential for utilising JQ1 and an autophagy inhibitor combination to effectively target CSCs in AML [192]. Despite this success, it remains challenging to specifically target only the pro-survival aspects of autophagy with respect to CSCs. For instance, autophagy induction can lead to the elimination of cancer cells or lead to cancer cell survival, in the early or late stages of cancer, respectively [27].

The regulatory network between miRNAs, lncRNAs, and the TME is also another angle that may be leveraged for therapeutic gain. For example, in chronic myelogenous leukaemia (CML), leukaemic stem cells (LSCs), may remain quiescent in the bone marrow, despite efforts to target BCR-ABL kinase activity by tyrosine kinase inhibitors (TKI). Resistance to TKI therapy in CML can occur due to LSCs quiescence induced by increased levels of *mir-126* [193]. It follows that the source of *miR-126* was Sca-1-negative endothelial cells of the bone marrow, which promoted quiescence and leukaemic growth. The knockdown or silencing of *miR-126* reduced leukaemia tumour-initiating capacity through engraftment of CML LSCs in the mouse host [193]. The mechanism underlying this regulation was shown to be the phosphorylation of SPRED1 by BCR-ABL that inhibited a modulator of *mir-126*, RAN/EXP-5/RCC1. Specifically, SPRED1 is a negative regulator of RAS proteins, while BCR-ABL phosphorylates SPRED1, allowing for the binding of this protein to RAN, which in turn interferes with *mir-126* shuttling and maturation mediated by RAN/EXP-5/RCC1 [193]. Hence the modulation of *mir-126* in combination with TKI therapy may allow for the effective targeting of quiescent LSCs in CML.

In a second study, the role of a tumour-suppressor, *MIR300,* in CML LSCs (CD34+ cells) was investigated. This miRNA displayed dose-dependent antiproliferative function, through CCND2/CDK6 inhibition and while it activated protein phosphatase 2A (PP2A), through SET inhibition, to induce apoptosis [194]. Further, *MIR300* expression led to expanded G0-G1 stem cell fraction, while its function was limited by an lncRNA, *TUG1. MIR300* was upregulated in CML LSCs as a means to induce quiescence, whereas BCR-ABL could downregulate *MIR300* in CML stem cells to inhibit growth arrest and apoptosis [194]. Interestingly, quiescent LSCs were able to selectively suppress the proapoptotic function of *MIR300* to prevent apoptosis, while allowing these cells to exit the cell cycle. The disruption of *TUG1-MIR300* interplay, could in turn lead to PP2A-dependent eradication of CML quiescent LSC in vitro and in mouse xenograft models. Hence, this *TUG1*/*miR-300*/PP2A regulatory network impairs LSC quiescence, thereby making this signalling pathway important for CML development and treatment [194].

Targeting lncRNAs may also have an unexpected advantage with respect to asymmetric division of CSCs, a process that may be regulated, in part, by lncRNAs. In evidence, the *lnc34a*, upregulated in colon CSCs, directly silenced *mir-34a* by recruiting DNMT3A and HDAC1 to methylate and deacetylate its promoter, independent of P53, the transcriptional regulator of this miRNA [195]. Subsequently, this led to an imbalance in *lnc34a* spatial dissemination, impacting CSC asymmetric divisions. This observation lends support to the targeting of CSC-specific properties such as asymmetric division and self-renewal, through the modulation of the associated lncRNAs [195]. Another example and a potential treatment strategy could be the modulation of *linc00617*, an lncRNA that binds to the promoter of SOX2, thus contributing to the self-renewal of CSCs in breast cancer [196].

An additional layer of complication occurring in parallel to these diverse events, includes clonal evolution of CSCs, akin to the premise of tumour evolution within a tumour, leading to heterogeneity within the CSC population [197]. CSCs gain genomic alterations that offer growth and survival advantages, receiving input during this process from both the TME, signalling pathways, and intrinsic regulatory networks. Heterogeneity can be detected within the CSC population in GBM, in which SOX2, OCT4, pSTAT3, KLF4, NANOG, and SALL4 expression defined CSC populations, while OCT4 levels varied between CSCs [42]. In evidence, Philadelphia-positive acute lymphoblastic leukaemia (ALL) primary patient tissue representing a single clinical entity with identifiable and recurrent genomic alterations, was used as a model to understand clonal evolution and in vivo growth. This study identified genetically diverse subclones within patient samples at diagnosis that varied in cancer-initiating cell counts and self-renewal capacity through in vivo growth in xenograft models [198], suggesting that eradicating all these subclones could prevent further tumour evolution [198]. It is therefore plausible that heterogeneity amongst CSCs, cancer-initiating cells, and persister cells, can impact therapy response, a process that can be tracked using next-generation sequencing approaches.

In conclusion, we endeavoured to delve deeper into understanding the characteristics of CSCs and methods used to identify and study these cells. We have provided specific examples to support the properties of CSCs in multiple solid and haematological cancers. We reviewed the implication of important cellular processes such as autophagy in the maintenance of CSCs and the current evidence supporting the roles that lncRNAs may play in CSC homeostasis alongside potential areas for therapeutic intervention. Using computational analysis, we have reinforced and substantiated the inherent links between these processes using protein–protein interactions and gene–disease association networks in this work. Finally, we have provided a unique view on a triangle of interlinked factors captured in this study inclusive of CSC properties, lncRNA, and autophagy.

## Figures and Tables

**Figure 1 cancers-13-01239-f001:**
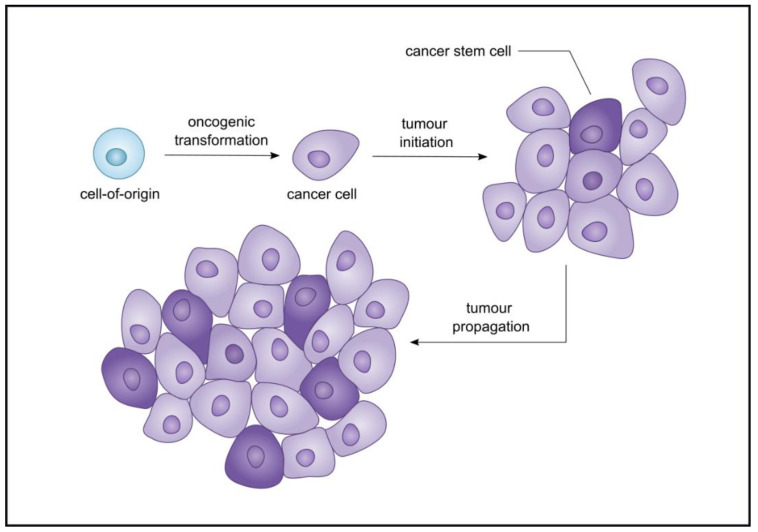
The distinction between cancer stem cells (CSCs) and cancer-initiating cells. A cancer-initiating cell (in blue) undergoes oncogenic transformation in order to develop a tumour, while a cancer stem cell (CSC, in dark purple) is not necessarily the transformed tissue-specific stem cell, but rather gives rise to the bulk of the tumour.

**Figure 2 cancers-13-01239-f002:**
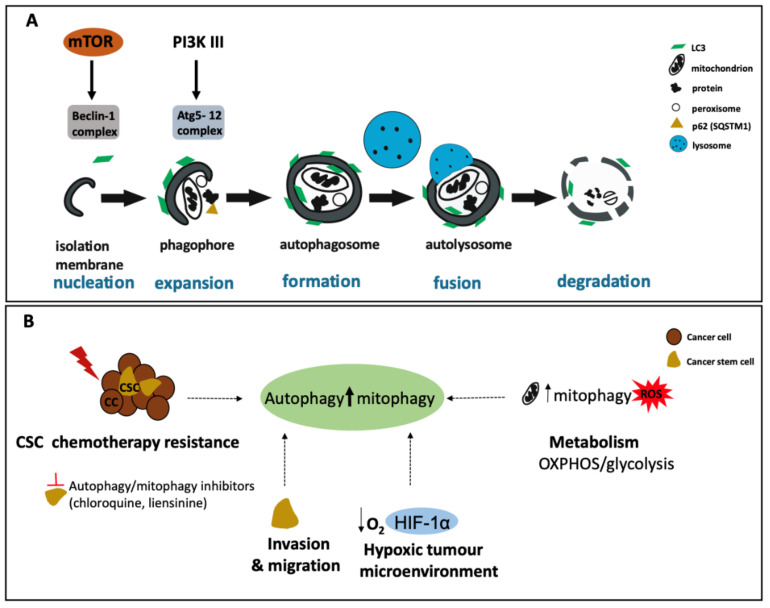
Autophagy, a pro-survival process, is elevated in CSCs. (**A**) The process of autophagy from nucleation to degradation. The process starts with the formation of an isolation membrane (regulated by the Beclin-1 complex), followed by the formation of a double membrane named the autophagosome which encapsulates cellular constituents for degradation. Following fusion with a lysosome, cell components in the autolysosome are degraded and recycled as nutrients and metabolites back into the cytosol. Several autophagy-related (Atg) proteins regulate different stages of autophagy (e.g., the Atg5-12 complex is vital for autophagosome formation). (**B**) Autophagy plays many roles in CSC survival. High levels of autophagy/mitophagy in CSCs promotes resistance to chemotherapy; potential treatments include autophagy/mitophagy inhibitors in combination with chemotherapy drugs. Induced autophagy stimulates CSC invasion and migration as well as survival in the hypoxic microenvironment. CSCs also depend on elevated mitophagy for their metabolism and to control ROS levels. Finally, CSCs show a mixed phenotype where both OXPHOS and glycolytic metabolism is employed in CSCs of different cancer models.

**Figure 3 cancers-13-01239-f003:**
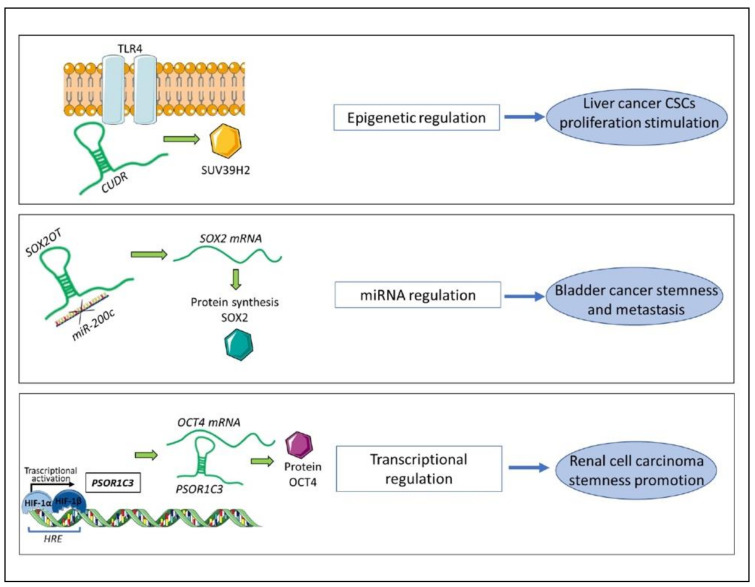
Long noncoding RNAs (LncRNAs) contribute to CSC features via the regulation of different cellular mechanisms. The three main mechanisms of actions of lncRNAs in CSCs regulation: Top panel: the lncRNA *CUDR* interacts with TLR4, thereby enhancing its function and promoting the activity of SUV39H2. SUV39H2 is involved in epigenetic regulation that stimulates liver cancer CSC proliferation. Centre panel: the lncRNA *SOX2OT* sequesters *miR-200c*, thereby inhibiting its function of targeting *SOX2* mRNA. Therefore, *SOX2OT* indirectly promotes translation of the stemness factor SOX2 in bladder cancer. Lower panel: the genetic locus of *PSORC1C3* contains an *HRE* responding to hypoxia-inducible factor (HIF) transcription factors. *HIF* triggers a CSC phenotype via *PSOR1C3* which interacts with *OCT4* mRNA, thereby stabilising it and promoting protein production and function in renal cell carcinoma cells. CSCs: cancer stem cells; HRE: hypoxia response element.

**Figure 4 cancers-13-01239-f004:**
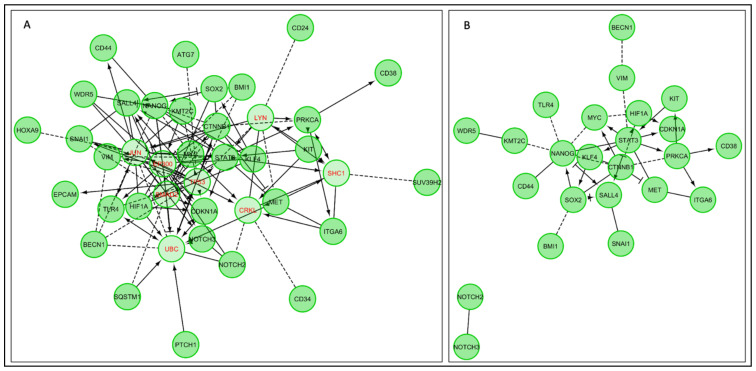
Reactome analysis of stemness proteins, CSC markers, autophagy genes, and lncRNAs discussed in this review (**A**) Reactome analyses of genes reviewed in this study and other genes that may provide a link between the genes discussed. (**B**) Reactome analyses performed using only genes reviewed in this study. Both analyses indicate functional interactions drawn from the Reactome database in Cytoscape. In this figure, edges indicate an interaction between two genes or their transcribed products. Dashed edges depict computationally predicted interactions with a probability of 0.95; solid edges indicate a shared protein complex or binding event. Arrowed edges indicate activation or catalytic activity, while blunted arrows indicate inhibition. Gene names in black are genes that were discussed in this review, while genes in red were not discussed in this review but were included because they provide a link between the genes discussed. Genes with no functional interactions are not shown.

**Figure 5 cancers-13-01239-f005:**
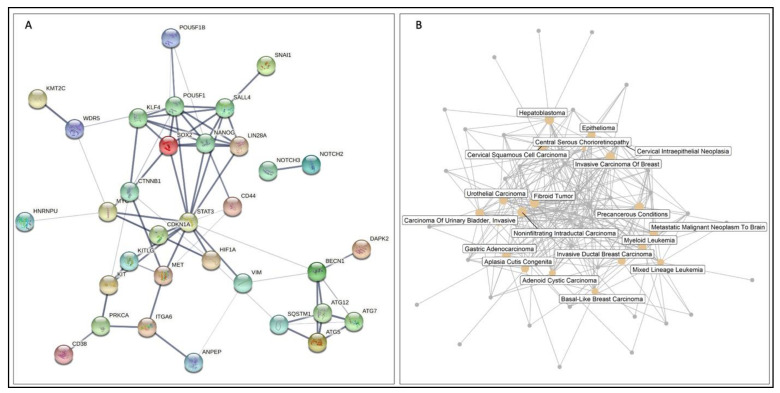
Functional and physical protein–protein interactions and DisGeNat analysis. (**A**) Physical and functional protein–protein interactions in the subset of genes that were reviewed in this work obtained using the STRING database v11.04. Nodes represent proteins, while the thickness of edges indicates the strength of the data obtained from a modified naïve Bayes algorithm. The evidence underlying the network was restricted to interactions in humans that were derived from curated databases (Biogrid), in addition to physical experiments, and co-expression (excluding text-mining, predicted, and orthologous evidence); 51 nodes, 70 interactions, 15 expected interactions, *p*-Value < 10^−16^. (**B**) A network generated by DisGeNet, depicting common diseases for which the network of genes was enriched. This information was obtained from the DisGeNet databases, using the ClusterProfiler package in R. The comparison includes a subset of 47 protein-coding genes discussed in this review that were curated in the DisGeNet database. The top 20 most enriched diseases were labelled, with the size of their corresponding gold node representing the adjusted −log10 *p*-Value of enrichment, while unlabelled grey nodes represent genes implicated in these cancer states. In total, 191 disease states were enriched (Bonferroni corrected *p*-Value < 0.005).

**Figure 6 cancers-13-01239-f006:**
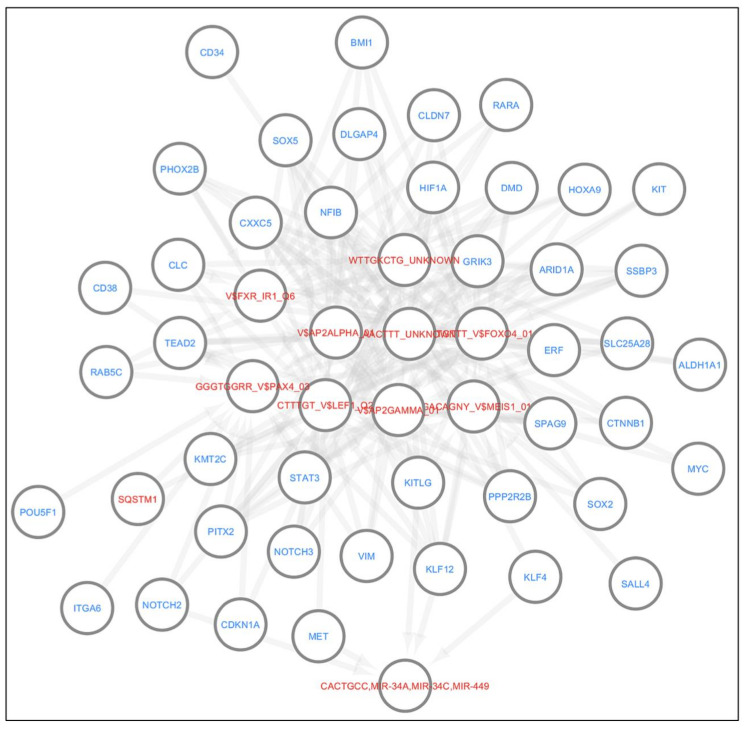
Genemania interrogation of transcription factor and miRNA targets. Transcription factor and miRNA target prediction from MSigDB using GeneMania. Targets are displayed in red, while genes are displayed in blue.

**Figure 7 cancers-13-01239-f007:**
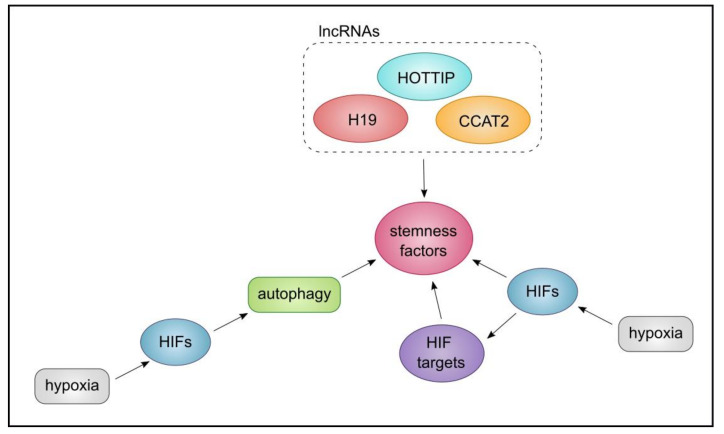
The contribution of autophagy and lncRNA to stemness factors in the tumour microenvironment (TME). In response to hypoxia, HIFs and their targets are triggered that can contribute to stemness factors. In turn, hypoxia-driven HIFs can trigger autophagy, as a means to survive the harsh conditions that and contribute to stemness factors. LncRNAs such as *H19*, *HOTTIP,* and *CCAT2,* could directly contribute to stemness factors.

**Figure 8 cancers-13-01239-f008:**
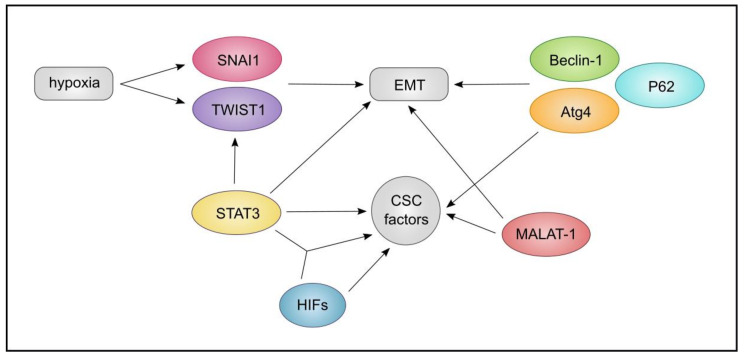
The interlinkage between the triangle of CSCs, lncRNAs, and autophagy with respect to epithelial-to-mesenchymal transition (EMT). Hypoxia in the TME can trigger SNAI1 and TWIST1 and relevant signalling pathways. STAT3 activation can contribute to EMT, while it can activate *TWIST1*. Further, STAT3 and HIFs directly or in combination can contribute to CSC factors. LncRNA *MALAT-1* can promote EMT and CSC factors. Autophagy mediators such as Beclin-1, P62, and Atg4 can contribute to both EMT and CSC factors.

**Table 1 cancers-13-01239-t001:** Examples of surface markers, stemness proteins, or factors that support the maintenance of stemness across multiple cancer types.

Cancer Type	Examples of Surface Markers, Stemness Proteins, or Factors Supporting the Maintenance of Stemness
AML	CD34+ CD38- or CD34- cells [17,28,29,30]
Oesophageal cancer	B7H4, LETM1, CD90 [31,32]
Colorectal cancer	CD44, CD133, ST6GALNAC1 [33]
Gastric cancer	CD44, SOX2, KLF4, and OCT4 [34,35]
Pancreatic cancer	CD133, CD24, CD44, ESA [36,37]
Liver cancer (hepatocellular carcinoma)	CD13, CD24, CD44, CD90, CD133 and EpCAM [38]
Lung Cancer (Non-small cell lung cancer)	SOX2 and NANOG [39] CDKN1A, SNAI1, and ITGA6 [40]
Glioblastoma multiforme	CD133 [41] SOX2, OCT4, NANOG, and SALL4 [42]
Osteosarcoma	SOX2 [43]
Breast cancer	CD44+ CD24-/low ALDH1+ [13,44]

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
