# Peer review of "The Role of Autophagy and lncRNAs in the Maintenance of Cancer Stem Cells"

_cancers, 2021, doi:10.3390/cancers13061239_

Round 1

Reviewer 1 Report

The aim of this review and its content are not correlated with the Title.

Actually the Authors discuss "the characteristics of CSCs identified in various cancer types and the role of autophagy and long noncoding RNAs (lncRNAs) in maintaining the homeostasis of CSCs. Further, they discuss methods to detect CSCs and strategies for treatment and relapse, taking into account the requirement to inhibit CSC growth and survival within the complex backdrop of cellular processes, microenvironmental interactions, and regulatory networks associated with cancer."

This means that the review is rather larger than what is written in the Title.

Actually so many reviews on either CSC identification, isolation and related methods have been already published. Moreover, methods to detect CSCs and strategies for treatment and relapse, taking into account microenvironment is a topic under research and several reviews, also in this case, have been published.

It is so strange that Authors like to enlarge the review and focus more to other topics than those highlighted in the Title.

The result is that the review is extremely dispersive and having so many topics becomes a book chapter rather than a true review.

Therefore the Authors must delete all the topics not included in the Title and focus on the two main topics mentioned in the Title (a bit forgotten).

Consequently the review will be much better to read. On the other hand so many unuseful citations will be deleted.

Regarding the two main topics the Authors must take into consideration that other more recent reviews and papers have been published, so they must up-to-date the whole.

For instancy, recent reviews and papers have been published on autophagy and tumors (i.e: Hypoxya-Mediated Autophagy in Tumor... Cancers (Basel) 2021 Jan 30;13(3):533.; The role of authophagy in...  Cancer Treat Rev. 2020 Aug;88:102043, as well as others).

Finally some pictures and a graphical abstract are needed.

Author Response

Reviewer 1:

The aim of this review and its content are not correlated with the Title. Actually, the Authors discuss "the characteristics of CSCs identified in various cancer types and the role of autophagy and long noncoding RNAs (lncRNAs) in maintaining the homeostasis of CSCs. Further, they discuss methods to detect CSCs and strategies for treatment and relapse, taking into account the requirement to inhibit CSC growth and survival within the complex backdrop of cellular processes, microenvironmental interactions, and regulatory networks associated with cancer." This means that the review is rather larger than what is written in the Title.

Actually so many reviews on either CSC identification, isolation and related methods have been already published. Moreover, methods to detect CSCs and strategies for treatment and relapse, taking into account microenvironment is a topic under research and several reviews, also in this case, have been published.It is so strange that Authors like to enlarge the review and focus more to other topics than those highlighted in the Title. The result is that the review is extremely dispersive and having so many topics becomes a book chapter rather than a true review. Therefore, the Authors must delete all the topics not included in the Title and focus on the two main topics mentioned in the Title (a bit forgotten). Consequentl, the review will be much better to read. On the other hand, so many unuseful citations will be deleted.

Regarding the two main topics the Authors must take into consideration that other more recent reviews and papers have been published, so they must up-to-date the whole. For instancy, recent reviews and papers have been published on autophagy and tumors (i.e: Hypoxya-Mediated Autophagy in Tumor... Cancers (Basel) 2021 Jan 30;13(3):533.; The role of authophagy in...  Cancer Treat Rev. 2020 Aug;88:102043, as well as others).

Finally some pictures and a graphical abstract are needed

-We thank the reviewer for these valuable comments, to which we have given our full consideration and regard. With sections 2 and 3 (CSCs properties and methods to detect them), before presenting the more novel topics pertaining to lncRNA and autophagy, we have provided an up-to-date view of these topics, to prepare the readers, of any level of expertise in this field, to familiarise themselves with the fundamental definitions and commonly used methods (e.g., colony formation, marker identification and in vivo tumour growth in 9 cancer types). With this clarification, we propose a resolution to the respectable reviewer. In accordance with the recommendation of the respectable reviewer and 2 other reviewers, and without any upper word limit imposed by MDPI, we have included more material in the autophagy section, and a more comprehensive linkage of CSCs, autophagy, and lncRNA in the new section 7, and have enhanced the focus of our discussion, as to further substantiate our story. Our revised paper now proposes a triangle of inherently linked, computationally reinforced factors inclusive of CSC properties/markers, lncRNA, and autophagy and their networks that through overlapping and non-overlapping mechanisms impact therapy response, hence the title of the paper is now directly linked to the core message. Further, our computational endeavours, which were based on genes collated from the entire manuscript, now more directly support the message of our revised paper. Finally, we critique strategies in the discussion to overcome this complex network for therapeutic gain. We would invite the reviewer to inspect these changes and sincerely hope that these changes are satisfactory.

Further, we have added a graphical abstract, a figure depicting differences between CSCs and TICs, and a triangle representation of CSCs, lncRNA, and autophagy with the view of EMT and CSCs factors to the revised paper, based on the recommendations of both reviewers 1 and 3.

We would also like to inform the reviewer, that we have conducted English language editing, as per request.

Reviewer 2 Report

The manuscript is interesting, novel and is well written and comprehensive. 
The only few critical points of concerns are as follow:

Comments:

  • About 20% of AML cases are characterized by absence of neoplastic CD34+ cells. In this case commonly small CD34+ (<1%) blast population does not contain Leukemia Stem Cells. By definition, these CD34- patients lack in CD34+CD38- and CD34+CD38+ leukemic populations. How do the authors comment this sentence about autophagy?
  • LSCs in AML have not been universally identified and it is highly likely that LSCs are quite heterogeneous and the phenotypic identification could be including different markers combinations depending on the AML subtype. Therefore, I would suggest to indicate the CD34+CD38- population not LSCs or differentiate with the current markers used.
  • The use of many abbreviations, making it difficult to read. I recommend including a list of all abbreviations used in the text and paying attention to write the full names of the acronyms reported in the text. 
  • References are few on lncRNA and more need to be added in the introduction and discussion sections. The authors should add the following references and discuss these two mirs ( miR-300 and miR-126) important for entrance or exit from quiescent state of leukemia quiescent cells in miRNA section and TUG-1 lncRNA should be discussed in the section of leukemia.

    (Silvestri G, Trotta R, Stramucci L, Ellis JJ, Harb JG, Neviani P, Wang S, Eisfeld AK, Walker CJ, Zhang B, Srutova K, Gambacorti-Passerini C, Pineda G, Jamieson CHM, Stagno F, Vigneri P, Nteliopoulos G, May PC, Reid AG, Garzon R, Roy DC, Moutuou MM, Guimond M, Hokland P, Deininger MW, Fitzgerald G, Harman C, Dazzi F, Milojkovic D, Apperley JF, Marcucci G, Qi J, Polakova KM, Zou Y, Fan X, Baer MR, Calabretta B, Perrotti D. Persistence of Drug-Resistant Leukemic Stem Cells and Impaired NK Cell Immunity in CML Patients Depend on MIR300 Antiproliferative and PP2A-Activating Functions. Blood Cancer Discov. 2020 Jul;1(1):48-67. doi: 10.1158/0008-5472.BCD-19-0039. PMID: 32974613; PMCID: PMC7510943).

    (Zhang B, Nguyen LXT, Li L, Zhao D, Kumar B, Wu H, Lin A, Pellicano F, Hopcroft L, Su YL, Copland M, Holyoake TL, Kuo CJ, Bhatia R, Snyder DS, Ali H, Stein AS, Brewer C, Wang H, McDonald T, Swiderski P, Troadec E, Chen CC, Dorrance A, Pullarkat V, Yuan YC, Perrotti D, Carlesso N, Forman SJ, Kortylewski M, Kuo YH, Marcucci G. Bone marrow niche trafficking of miR-126 controls the self-renewal of leukemia stem cells in chronic myelogenous leukemia. Nat Med. 2018 May;24(4):450-462. doi: 10.1038/nm.4499. Epub 2018 Mar 5. PMID: 29505034; PMCID: PMC5965294).

Author Response

Reviewer 2:

Comments and Suggestions for Authors

The manuscript is interesting, novel, and is well-written, and comprehensive. 

We thank the respectable reviewer for this comment.

The only few critical points of concerns are as follow:

Comments:

About 20% of AML cases are characterized by the absence of neoplastic CD34+ cells. In this case commonly small CD34+ (<1%) blast population does not contain Leukemia Stem Cells. By definition, these CD34- patients lack in CD34+CD38- and CD34+CD38+ leukemic populations. How do the authors comment this sentence about autophagy?  In the discussion, we have added a section about LSCs (CD43+ CD38-) and autophagy to reflect on this process in AML. In AML, the combination therapy of BET and autophagy inhibitors has led to more effective targeting of LSCs. Hence LSCs are utilising autophagy to survive drug treatment in this cancer (PMID:28118076). However, to our knowledge, there has not been extensive investigation of the role of autophagy in the CD34- population that the respectable reviewer is referring to, however, we would predict that CD34- cells, as cells with self-renewal capacity should, in principle, rely on pro-survival processes to survive harsh changes to the microenvironment.

LSCs in AML have not been universally identified and it is highly likely that LSCs are quite heterogeneous and the phenotypic identification could be including different markers combinations depending on the AML subtype. Therefore, I would suggest to indicate the CD34+CD38- population not LSCs or differentiate with the current markers used. We thank the reviewer for this valuable insight and updated view of AML CSC. We have reflected this view in the leukaemia review section and have inserted relevant papers (PMID: 20053758, PMID:8630378). Also, table 1, reflects this updated view.

The use of many abbreviations, making it difficult to read. I recommend including a list of all abbreviations used in the text and paying attention to write the full names of the acronyms reported in the text. 

We thank the reviewer and have inserted an abbreviation table at the end of the manuscript to further assist our readers with abbreviations.

References are few on lncRNA and more need to be added in the introduction and discussion sections. The authors should add the following references and discuss these two mirs (miR-300 and miR-126) important for entrance or exit from quiescent state of leukemia quiescent cells in miRNA section and TUG-1 lncRNA should be discussed in the section of leukemia.

 We have elaborated on the papers kindly suggested by the reviewer (PMID: 32974613, PMID: 29505034) in the discussion since we have appraised specific targeting of CSCs by various methods, in this section that was a better fit for the theme of these papers.  We thank the reviewer for these valuable suggestions. In addition, we have added PMID:27077950 to the discussion (signposted).

We have added a new section 7 to link CSC characteristics, lncRNA, and autophagy, which also features more references on lncRNA that features papers including PMID: 30218296 and PMID: 28187439. With respect to the introduction, we have added a paragraph to better introduce lncRNA and autophagy featuring PMID:25842979, PMID:33291403, PMID: 24829860, and PMID:30218296. We would like to point out, that we anticipate that at least 65 references of the 223 references used in this review are directly related to lncRNA. We sincerely hope that the respectable reviewer finds these improvements satisfactory.

Reviewer 3 Report

The story of Jahangiri et al. is a comprehensive review on the properties/potential/significance  of tumor stem cells, with the emphasis on their consequences for tumor homeostasis under chemotherapeutic stress and the role autophagy/lncRNAs in the regulation of tumor stem cells phenotype. The paper is well written and informative; as such it deserves the publication in Cancers. Below, I enclose a few comments that can potentially help to further improve/strengthen its message: 

- the distinction between tumor initiating cells and tumor stem cells is a matter of many misunderstandings. As the Authors comprehensibly cover this point, I would suggest the introduction of the diagram that would depict the differences between TICs and CSCs;

- the concept of "clonal evolution of CSCs" should be emphasized as a consequence of CSC drug-resistance/heterogeneity;

- since the role of autophagy and lncRNAs in CSC phenotype is mentioned in the title/abstract of the paper, I would expect a separate chapter containing a more detailed/focused (even if partly speculative) description of the mutual links within "a triangle" defined by (i) CSC phenotype/potential, (ii) autophagy and (iii) lncRNA (perhaps partly mediated by miRNAs). A more comprehensive notion on the possible significance of these interrelations would also be desirable for the chapter (7; Conclusion).

Author Response

Reviewer 3:

Comments and Suggestions for Authors

The story of Jahangiri et al. is a comprehensive review on the properties/potential/significance of tumor stem cells, with the emphasis on their consequences for tumor homeostasis under chemotherapeutic stress and the role autophagy/lncRNAs in the regulation of tumor stem cells phenotype. The paper is well written and informative; as such it deserves the publication in Cancers. Below, I enclose a few comments that can potentially help to further improve/strengthen its message: 

We thank the respectable reviewer for these comments.

The distinction between tumor-initiating cells and tumor stem cells is a matter of many misunderstandings. As the Authors comprehensibly cover this point, I would suggest the introduction of the diagram that would depict the differences between TICs and CSCs;

We have added figure 1, to support the text aimed at distinguishing between TICs, and CSCs.

The concept of "clonal evolution of CSCs" should be emphasized as a consequence of CSC drug-resistance/heterogeneity;

We have added a paragraph in the discussion (highlighted) to expand upon this notion.

Since the role of autophagy and lncRNAs in CSC phenotype is mentioned in the title/abstract of the paper, I would expect a separate chapter containing a more detailed/focused (even if partly speculative) description of the mutual links within "a triangle" defined by (i) CSC phenotype/potential, (ii) autophagy and (iii) lncRNA (perhaps partly mediated by miRNAs). A more comprehensive notion on the possible significance of these interrelations would also be desirable for the chapter (7; Conclusion).

We have provided a new section 7 to substantiate the proposed triangle and elaboration of the discussion with a view of leveraging this triangle to more effectively target CSCs. We would like to note that lncRNA- miRNA links have been also featured in the discussion.

Round 2

Reviewer 1 Report

The Authors did not understand in full what this reviewer wrote in the previous comments.

Having them added instead of deleting some parts in this review it is extremely long.

Moreover, as this review previously wrote, it cannot include so much news, tough the Authors tried to link each other.

Something can be included, but not pages and pages.

In my previous comments I asked in particular the following:

" the Authors discuss the characteristics of CSCs identified in various cancer types and the role of autophagy and long noncoding RNAs (lncRNAs) in maintaining the homeostasis of CSCs. Further, they discuss methods to detect CSCs and strategies for treatment and relapse, taking into account the requirement to inhibit CSC growth and survival within the complex backdrop of cellular processes, microenvironmental interactions, and regulatory networks associated with cancer."

Now Authors ONLY changed the Title and added some sentences or paragraphs to link the parts, while they had to chose which part to delete.

I also wrote: "Actually so many reviews on either CSC identification, isolation and related methods have been already published. Moreover, methods to detect CSCs and strategies for treatment and relapse, taking into account microenvironment is a topic under research and several reviews, also in this case, have been published! This means that all the above must be deleted!

Moreover I wrote " Therefore, the Authors must delete all the topics not included in the Title and focus on the two main topics mentioned in the Title" : means deleting the others and NOT adding!

Therefore, so many unuseful citations will be deleted!

In summary :The Authors have performed some of the changes requested by this reviewer, however they are still missing the most important point: the manuscript is too long and many part are redundant or not necessary. In details the introduction must be shortened, the section regarding the methods to characterize CSCs as well as the sections describing the role of CSCs in different tumors are not necessary and should be removed or shrunk into a section no longer than few lines.

I appreciate the efforts, but a review cannot be so long. The concise reviews are much more readable than longer ones.

Author Response

We thank the reviewer for providing clarity to their comments and would like to inform them that we have performed significant trimming of the manuscript (5 pages less content), in a manner that is not detrimental to an understanding of the later sections. We have enclosed the revised manuscript and would invite the respected reviewer to review these changes in light of the comments below with regards to the suggested deletion of sections 2-3 and significant shortening of the introduction:

1- Deletion of sections 2-3 would void table 1 and CSC genes described therein. This would lead to the entire computational section (section 6) inclusive of 3 figures (4-6) to be lost, since the genes in table 1 are integral to the computational networks found. If we delete table 1, we will not be able to explain why genes such as ITGA6 and LETM1 were selected for further discussion. If we keep the table, it would remain as a stand-alone unexplainable entry. Also, the new information concerning AML CSC markers requested by reviewer 2, will also be lost.

2- Deletion of sections 2-3 would result in the triangle section (section 7 added due to the request of reviewer 3), appearing superficial, since one angle of this triangle (CSC properties) would no longer exist.

3- Significant shortening of the introduction as suggested could lead to the loss of figure 1 (added due to the request of reviewer 3).

In light of these ramifications, we hope that the reviewer appreciates the reduction of content performed, while also preserving later sections.

Reviewer 3 Report

I have no more comments

Author Response

Many thanks

Round 3

Reviewer 1 Report

The Authors provided their third version in a too fast way: in a very few days. This leads to having corrected the text in a hurry leaving several parts still not well organised and long.

It is ameliorated, but not as required.